# FPLV: Enhancing recommender systems with fuzzy preference, vector similarity, and user community for rating prediction

Zhan Su◉*, Haochuan Yang◉, Jun Ai🅿◉*

School of Optical-Electrical and Computer Engineering, University of Shanghai for Science and Technology, Shanghai, P.R.China

◉ These authors contributed equally to this work.
* suzhan@usst.edu.cn (ZS); aijun@usst.edu.cn (JA)

**Data Availability Statement:** Data relevant to this study are available from GitHub at https://github.com/PlayerAI/PONE-D-22-28030.

**Funding:** Zhan Su received supported by the National Natural Science Foundation of China

## Abstract

Rating prediction is crucial in recommender systems as it enables personalized recommendations based on different models and techniques, making it of significant theoretical importance and practical value. However, presenting these recommendations in the form of lists raises the challenge of improving the list's quality, making it a prominent research topic. This study focuses on enhancing the ranking quality of recommended items in user lists while ensuring interpretability. It introduces fuzzy membership functions to measure user attributes on a multi-dimensional item label vector and calculates user similarity based on these features for prediction and recommendation. Additionally, the user similarity network is modeled to extract community information, leading to the design of a set of corresponding recommendation algorithms. Experimental results on two commonly used datasets demonstrate the effectiveness of the proposed algorithm in enhancing list ranking quality, reducing prediction errors, and maintaining recommendation diversity and accurate user preference classification. This research highlights the potential of integrating heuristic methods with complex network theory and fuzzy techniques to enhance recommendation system performance with interpretability in mind.

## Introduction

In recent years, the rise of information overload has prompted the development of various recommender systems, offering users a means to effectively filter vast amounts of data [1]. At its core, a recommender system aims to prioritize information based on individual user preferences, retaining and prioritizing content that is relevant or anticipated to be of interest to the user. This approach mitigates the challenges posed by information overload, sparing users from expending excessive time and energy when confronted with an abundance of data [2].

The recommender system operates on two primary sources of information: the "footprint" generated from users' online activities and interactions, as well as insights derived from the interactions between the target user and other users, or the evaluations of the target item by other users [3].

(Grant No. 61803264). Their website is at https://www.nsfc.gov.cn/. The funders had no role in study design, data collection and analysis, decision to publish, or preparation of the manuscript.

**Competing interests:** The authors have declared that no competing interests exist.

Recommender systems, established in the early 1990s, are now widely used across diverse fields. Major websites like YouTube and Netflix use them to recommend personalized entertainment content [4, 5]. E-commerce companies, including Amazon, utilize them to offer accurate product recommendations for enhanced competitiveness [6]. Social networks like Twitter and Spotify have also developed their own recommender systems [7, 8]. These systems play a crucial role in enhancing user experiences and making tailored recommendations.

Currently, mainstream recommender systems employ three types of recommendation algorithms based on their working methods [9, 10]:

- Content-based recommendation algorithm (CB)

- Collaborative filtering recommendation algorithm (CF)

- Hybrid recommendation algorithm

Content-based recommendation algorithms focus solely on user characteristics and item attributes [11]. For instance, a movie website utilizing a content-based approach will base its movie recommendations on each user's viewing history and search preferences, suggesting movies similar to their past interactions.

The collaborative filtering (CF) recommendation algorithm utilizes a rating matrix, incorporating user's community neighbors and their item ratings [11]. The critical step in CF involves computing similarity between users or items [12]. Unlike content-based approaches, CF introduces a "neighborhood" concept, which facilitates recommendations based on community ratings. This not only suggests items users are likely to enjoy but also predicts potential interests. Notably, many Matrix Factorization and Deep Learning algorithms fall under CF, as they employ known community ratings to predict unknown ones.

The hybrid recommendation algorithm is a fusion approach that combines the advantages of CB and CF [13]. Researchers sought to merge these distinct algorithms to harness their strengths simultaneously. Mainstream hybrid techniques can be categorized into seven types: weighting, selection, mixing, feature combination, cascade, feature enhancement, and meta-level [14]. These hybrid algorithms cater to diverse application scenarios and may incorporate additional machine learning algorithms, leading to complex models with potentially reduced generalization performance.

Recommender systems, while immensely beneficial, currently confront several challenges. Some key challenges include improving the ranking quality of recommended items, dealing with the cold start problem, overcoming data sparsity poses another hurdle, ensuring fairness and avoiding algorithmic biases, as well as the concern of privacy and security issues.

Recent years have witnessed significant advancements made by scientists and engineers in their relentless pursuit of enhancing system performance. Collaborative filtering algorithms, owing to their high interpretability, ease of implementation, training, and generalization, continue to be a prominent and fervently researched topic in the field of recommender systems [15].

Various competitive similarity measures continue to be proposed due to the pivotal role similarity plays in collaborative filtering (CF) algorithms [16], including quasi-norm-based sub-similarity [17] and resonance similarity (RES) [18]. Additionally, classic similarity algorithms and their improved versions, such as Cosine similarity [19] and distance-weighted cosine similarity metric [20], as well as Pearson similarity [21, 22] and its enhancement based on item frequency [22], also contribute to the growing array of similarity options for CF algorithms [9].

Apart from enhancing recommendation and prediction through improved similarity computation, the integration of aids and models based on diverse theories also contributes significantly to improving the performance of recommender systems.

Matrix Factorization (MF) recommendation algorithms [23–25] have gained widespread popularity and success in the field of recommender systems. These algorithms aim to decompose the user-item interaction matrix into latent factors, effectively representing users and items in a lower-dimensional space. By learning these latent factors, MF algorithms can capture the underlying patterns and relationships between users and items, enabling accurate prediction of missing ratings and generating personalized recommendations. The popularity of MF algorithms is attributed to their ability to handle data sparsity, address the cold start problem, and offer improved recommendation accuracy in various application domains.

Deep learning (DL) recommendation algorithms [26, 27] have emerged as a cutting-edge approach in the realm of recommender systems. By utilizing multiple layers of interconnected neurons, deep learning models usually encode user and item as embeddings, effectively capture high-level features from raw input data, and decode output as accurate and sophisticated recommendations. Deep learning recommendation algorithms excel at handling unstructured data, handling sequential user behaviors, and adapting to various types of recommendation tasks. Their ability to leverage vast amounts of data and learn intricate representations makes them particularly adept at generating personalized and context-aware recommendations, advancing the state-of-the-art in the field of recommender systems.

However, the training process of Matrix Factorization (MF) and Deep Learning (DL) methods in recommender systems is slow and resource-intensive [23]. MF algorithms decompose large user-item interaction matrices, demanding significant computational effort, particularly for datasets with numerous users and items. As a result, MF algorithms in large-scale recommender systems may face scalability issues. Additionally, MF methods lack the capability to incorporate contextual information or side features, limiting their ability to capture complex user preferences and item characteristics.

Similarly, DL recommendation algorithms [26], with their complex neural network architectures, often demand extensive training time and substantial computing resources to optimize the vast number of model parameters. Moreover, while DL methods achieve impressive predictive performance, one significant drawback is their lack of explainability. The intricate nature of deep neural networks makes it challenging to interpret how the model arrives at its recommendations, hindering the ability to provide transparent and understandable reasoning for users. Furthermore, while DL methods excel in certain domains, their generalizability across different areas of recommender systems remains a concern.

Given the slow and resource-intensive training process of MF and DL methods in recommender systems, along with their scalability challenges and limitations in incorporating contextual information, our work aims to explore alternative theories and methods to enhance RS performance without relying on MF and DL approaches.

For example, the user's rating of items forms the foundation for most similarity and MF algorithms, directly influencing the prediction accuracy of each recommender system. Yet, users often express fuzzy and uncertain feelings, such as like, dislike, general, or no feeling, making precise ratings challenging. These fuzzy sentiments, less subjectively influenced by users than exact scores, help improve user similarity calculation, neighbor selection, and reduce prediction errors. Consequently, effectively handling fuzzy user feedback becomes a vital step in similarity algorithms.

Fuzzy logic focuses on managing concepts, objects, or information that lacks precise representation in the real world [28]. It primarily relies on fuzzy sets [29] and membership functions [30]. The membership function maps elements in the research scope to corresponding categories through varying membership values. A fuzzy set encompasses all categories, including combinations of each pair of elements in each category and their membership values.

The recommender system that adopts fuzzy logic similarity algorithm is called fuzzy logic recommender system [31], which is generally divided into two types: content-based fuzzy logic recommender system and collaborative filtering fuzzy logic recommender system.

Shojaei and Saneifar [32] introduced a new multi-level fuzzy similarity measure (MFSR) for recommender systems, incorporating popularity and saliency. They proposed a similarity computation hierarchy to enhance recommendation accuracy and quality. Experimental evaluation demonstrated that their multi-level fuzzy similarity algorithm, combined with fuzzy logic, outperformed baseline algorithms PIP and NHSM in MAE, F1, recall, and accuracy. This highlights fuzzy logic's effectiveness in addressing uncertainty and identifying ambiguity when measuring item-user similarity.

In collaborative filtering, fuzzy logic primarily extends typical similarity measures (e.g., Pearson and Cosine [10]) to their corresponding fuzzy concepts [28]. Houshmand-Nanehkaran et al. [33], Bouacha and Bekhouche et al. [34], and Surya Kant et al. [35] have applied fuzzy logic in collaborative filtering by transforming user's item ratings into fuzzy preferences, which then replace ratings for subsequent calculations. Fuzzy preference denotes the subjective evaluation, linguistically converted from objective scores through fuzzy logic, often represented in phrase form.

Various algorithms adopt diverse approaches to define fuzzy preferences, but ultimately, they all employ preference sets as fuzzy sets. For instance, Zhang et al.'s method [36] introduces five types of fuzzy sets: strongly interested (SI), more interested (MI), interested (I), less interested (LI), and not interested (NI). Utilizing this fuzzy set, they propose a fuzzy Pearson correlation coefficient method to calculate user and item similarity.

The aforementioned reasons underscore our motivation to explore and implement fuzzy-related methods to enhance the performance of recommender systems.

On the other hand, researchers have demonstrated that the structural information within recommender systems (RS) can be effectively leveraged to enhance prediction and recommendation capabilities. Ai et al. [37] propose modeling users and items as a user-user or item-item network based on their similarity, revealing valuable structure information for improved prediction and recommendation. Additionally, centrality measures [38] and community detection [39] in the similarity network can further enhance RS performance. To avoid the computational complexity of community detection, Ai et al. suggest utilizing K-core decomposition to cluster users in the similarity network [40]. Given these findings, it is justifiable to incorporate network features in our research to enhance the overall performance of RS.

Similarly, several other methods have incorporated multi-dimensional data into prediction. Zhang et al. [41] propose a novel ensemble approach of Markov chains and complex networks for clothing recommendation. Li et al. [42] introduce a combined method of LDA and Word2-vec to extract joint features, leveraging various data types like product attributes, user reviews, and friend relationships to enhance the accuracy of recommendation algorithms. These approaches exemplify the efforts to utilize diverse data sources and advanced techniques to optimize the recommendation process.

In the field of recommender systems, various other methods are employed to enhance prediction and recommendation performance. These include utilizing users' diverse behaviors for similarity [43], employing normalization, dimensionality reduction, and classification techniques [44], as well as resource allocation strategies [45]. Each of these approaches contributes to the continuous evolution and improvement of recommender systems, catering to different aspects and requirements of the recommendation process.

Significantly, Su et al. [46] introduced a method to capture user similarity across various item types. Unlike traditional CF algorithms that consider overall inter-user similarity, this method scales the inter-user similarity values into a similarity vector, reflecting similarity

across different item classifications. For instance, two users may exhibit high similarity in science fiction films but have markedly different and low similarity in their preferences for drama and romance films. By constructing this vector similarity of users, the experiments demonstrated notable enhancements in several prediction and recommendation metrics. This approach offers a more nuanced and accurate representation of user preferences, leading to improved recommendation performance. In addition, there is research in the field of DL that considers label information to enhance RS performance [47].

Our research aims to tackle the challenge of enhancing the ranking of recommendation lists by exploring a novel approach that combines fuzzy logic with vector similarity methods. While previous studies have independently investigated the potential of these two methodologies in recommender systems, our work seeks to integrate them for a more comprehensive and accurate representation of user preferences. By incorporating fuzzy membership functions to handle user preference uncertainty and leveraging vector similarity to capture intricate user-user relationships, our proposed method strives to provide improved ranking accuracy and recommendation quality.

Therefore, the main contributions of this paper can be summarized as:

1. First, we combine the fuzzy preference with the item label to generate the user's fuzzy preference label vector for items, effectively capturing users' uncertain and imprecise preferences.

2. Secondly, we design a group of methods based on the similarity measurement of fuzzy preference label vector, with one method being parameter-free, enhancing the accuracy of similarity computations in recommender systems.

3. Finally, based on the user-user similarity network, our method utilizes different information characterized from RS and alleviates the conflict between prediction accuracy and recommendation diversity.

## Methodology

### Related works

Among many similarity measurement methods, Cosine similarity is a simple and direct algorithm with excellent effect [48], its definition is shown in the Eq 1. The value of *CosSim* ranges from [0, 1] when there is no negative rating, and the larger the value, the higher the user similarity.

$$CosSim(q, w) = \frac{\sum_{i=1}^{n}(r_{qi})(r_{wi})}{\sqrt{\sum_{i=1}^{n}(r_{qi})^2}\sqrt{\sum_{i=1}^{n}(r_{wi})^2}} \tag{1}$$

where $r_{qi}$ and $r_{wi}$ represent the user $q$ and $w$'s respective ratings of the item $i$, $i \in I_{q,w}$, and $I_{q,w}$ is the set of items jointly rated by users $q$ and $w$.

### Method overview

As depicted in Fig 1, the prediction algorithm presented in this paper comprises three key steps: ratings fuzzification, item vectorization, and the establishment of the link prediction algorithm. In the first step, a fuzzy membership function is proposed to transform the user's item ratings into fuzzy preferences of "like" (Like) and "dislike" (Dislike). This conversion yields a binary fuzzy set, enhancing the effectiveness of collaborative filtering. Next, the paper expands items into item label vectors with labels as elements, combining them with fuzzy

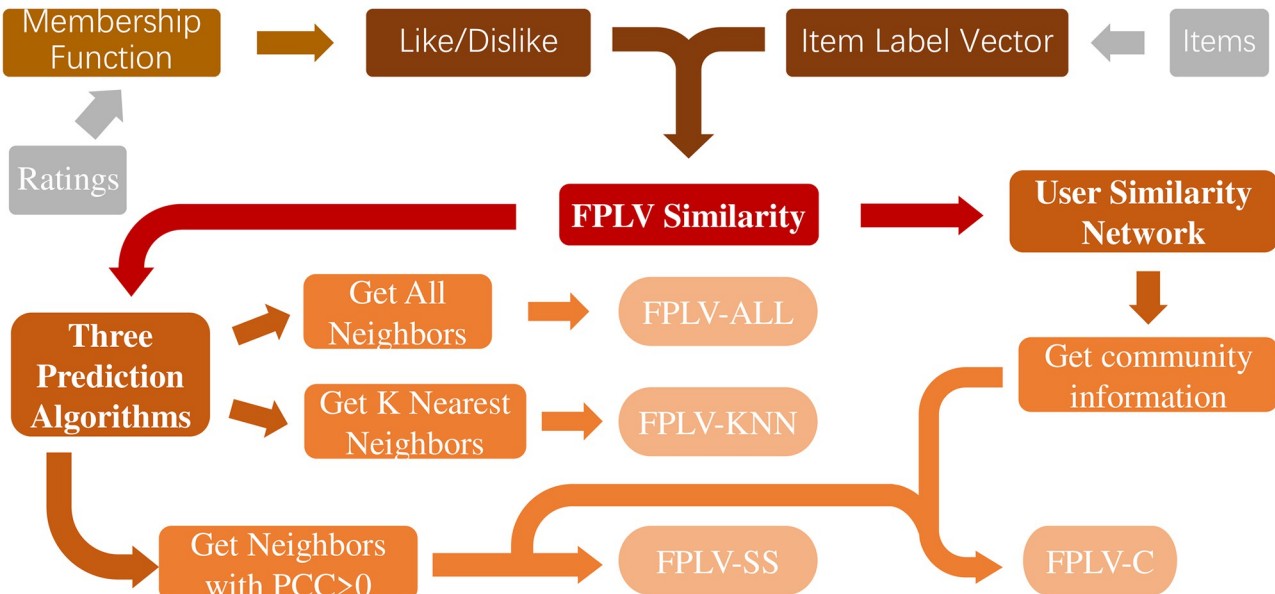

**Fig 1. Flowchart of the algorithm.** Items and ratings serve as the input information for the algorithm, and the prediction results are generated through FPLV-ALL, FPLV-KNN, FPLV-SS, and FPLV-C methods.

preferences to introduce new similarity calculation and recommendation equations. In the final step, three prediction algorithms, namely fuzzy-preference label vector CF with all qualified neighbors (FPLV-ALL), fuzzy-preference label vector CF with *K* nearest neighbors (FPLV-KNN), and fuzzy-preference label vector CF with similarity selection [49] (FPLV-SS), are proposed based on the similarity of fuzzy preference label vectors. A user-user similarity network is then established using the proposed similarity method, and fuzzy-preference label vector CF with community characteristics (FPLV-C) is developed based on the results of community division. These steps collectively contribute to an innovative and comprehensive prediction approach in our research.

The algorithm's general flow is as follows:

1. Obtain the fuzzy preference index of each user using the fuzzy preference algorithm.

2. Generate the original label vector of the item.

3. Compute the like index similarity and dislike index similarity based on the fuzzy preference index among users and the original label vector of the item.

4. Calculate the average of the two similarities as the similarity between the two users.

5. Establish the user similarity network with users as nodes and similarity as edges.

6. Utilize the community detection algorithm to obtain community information from the similarity network.

7. Combine community information for user-item prediction.

The general process structure of this algorithm is as follows:

## Ratings fuzzification

In this step, we analyze users in the rating matrix individually to obtain each target user's fuzzy preference for each item. Using a fuzzy membership function inspired by fuzzy mathematics, we calculate the corresponding like index $fp^{\text{like}}$ or dislike index $fp^{\text{dislike}}$. The application of the fuzzy membership function enhances the objectivity, logic, and accuracy of user similarity calculation, thereby reducing the error in final score predictions.

We categorize the user's fuzzy preference for items as "like" and "dislike". Comparing each user's item ratings with their respective rating averages, we determine their fuzzy preference. If the user's rating for an item is higher than its average rating, we consider it a "like"; otherwise, it is a "dislike". Fig 2 illustrates the preference relationships between users and items. The orange solid line and green solid line represent "like" preferences, while the orange dashed line and green dotted line indicate "dislike" preferences. For instance, users $w$ and $e$ like items 2, 3, and 4, but dislike items 1 and 5. Similarly, users $q$ and $r$ both like items 2, 3, and 4 but dislike items 1 and 5. Based on these fuzzy preferences, we can determine the similarity between users $w$ and $e$ as the highest among the four users, and similarly, the similarity between users $q$ and $r$ is the highest among them.

After getting the user's preference for the item, apply the Eq 3 and the Eq 4 to get the user's fuzzy preference index for the item.

$$R: \qquad \text{IF} \quad r_{qi} > \bar{r}_q,$$
$$\text{THEN} \quad fp^{\text{like}} = \mu_1(r_{qi}) \quad \text{and} \quad fp^{\text{dislike}} = 0, \tag{2}$$
$$\text{ELSE} \quad fp^{\text{like}} = 0 \quad \text{and} \quad fp^{\text{dislike}} = \mu_2(r_{qi})$$

$$\mu_1\left(r_{qi}\right) = \begin{cases} 1, & r_{qi} = r_q^{\text{max}} \\ \dfrac{r_{qi} - \bar{r}_q}{r_q^{\text{max}} - \bar{r}_q}, & \bar{r}_q \le r_{qi} < r_q^{\text{max}} \\ 0, & r_{qi} < \bar{r}_q \end{cases} \tag{3}$$

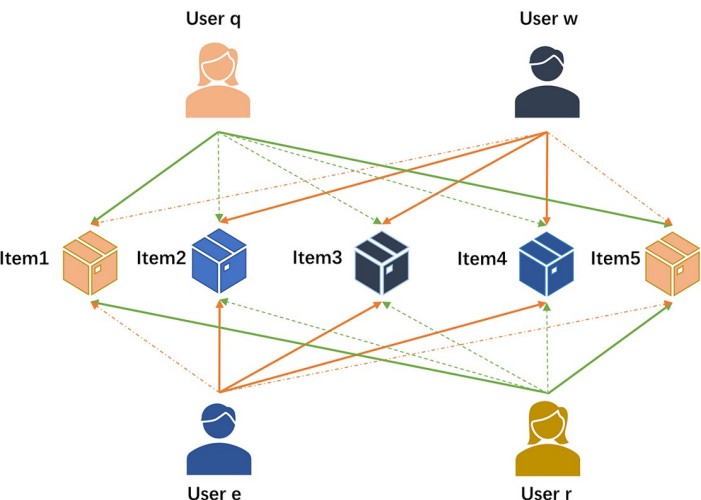

**Fig 2. User preferences for 5 items.** Solid lines represent "like" preferences, while dotted lines indicate "dislike" preferences. It highlights that the similarity between users $w$ and $e$ is higher than that of user $w$ with others, and similarly, the similarity between users $q$ and $r$ is higher than that of user $q$ with others.

$$\mu_2\left(r_{qi}\right) = \begin{cases} 1, & r_{qi} = r_q^{\min} \\ \dfrac{\bar{r}_q - r_{qi}}{\bar{r}_q - r_q^{\min}}, & r_q^{\max} \leq r_{qi} < \bar{r}_q \\ 0, & r_{qi} > \bar{r}_q \end{cases} \quad (4)$$

where $r_{qi}$ denotes the rating of item $i$ by user $q$, $\bar{r}_q$ represents the average rating given by user $q$, while $r_q^{\min}$ and $r_q^{\max}$ represent the lowest and highest scores ever rated by user $q$, respectively. $\mu_1(r_{qi})$ represents user $q$'s fuzzy preference index for item $i$, while $\mu_2(r_{qi})$ denotes the dislike index. As a user's preference for an item can only be either "like" or "dislike", $\mu_1(r_{qi})$ and $\mu_2(r_{qi})$ are mutually exclusive and will have a value of 0 depending on the user's preference for the item.

After obtaining the user's fuzzy preference index for the item, the next challenge is to combine this index with the item's label vector to generate the fuzzy preference label vector.

## Item vectorization

In recommender system research, similarity calculation methods significantly impact the accuracy of target user rating prediction and recommendation results. Previous research on fuzzy preference has averaged the Pearson similarity value of the like index and dislike index for jointly rated items by two users to obtain user similarity. However, this approach treats user similarity as a scalar and disregards the impact of differences in the detailed attributes of the items on the calculation of user similarity.

Consider two users, $p$ and $q$, both displaying a vague preference for 'like' towards sports videos. Based on previous research, the recommender system would consider these users highly similar. However, when predicting a video about a specific sport, the preferences of users $p$ and $q$ may diverge due to the detailed properties of the video, which go beyond the broad category of sports. If the previous algorithm is used for rating prediction and recommendation, the results may have a significant deviation.

The proposed user fuzzy preference similarity algorithm, based on item label vectors, aims to leverage rating and item information more effectively to address rating prediction and recommendation challenges for items with detailed attributes. This approach yields improved results using only a small number of neighbors for similarity computation. To keep the algorithm's complexity manageable, we directly obtain all item properties from the dataset, avoiding the need for additional algorithms. Item attributes serve as labels, and assuming there are $k$ types of labels in the dataset, we generate a $k$-dimensional label vector $L$ containing all labels, i.e., $L = \{l_1, l_2, \ldots, l_t, \ldots, l_k\}$.

Fig 3 shows a schematic diagram of item label vectors. There are $N$ items, each with a label vector of length $k$. Not all items contain all labels, so in the label vector $L$, we set the elements corresponding to labels present in the item to 1 and the rest to 0, thus generating the original label vectors for all items.

## Similarity based on fuzzy-preference label vector

After obtaining the item label vectors, we propose a new collaborative filtering algorithm. We create a similarity vector that adapts to the item label vector and further divide the user similarity into like and dislike, effectively utilizing the calculated fuzzy preference index. Finally, we average the two similarities to obtain the final similarity between users.

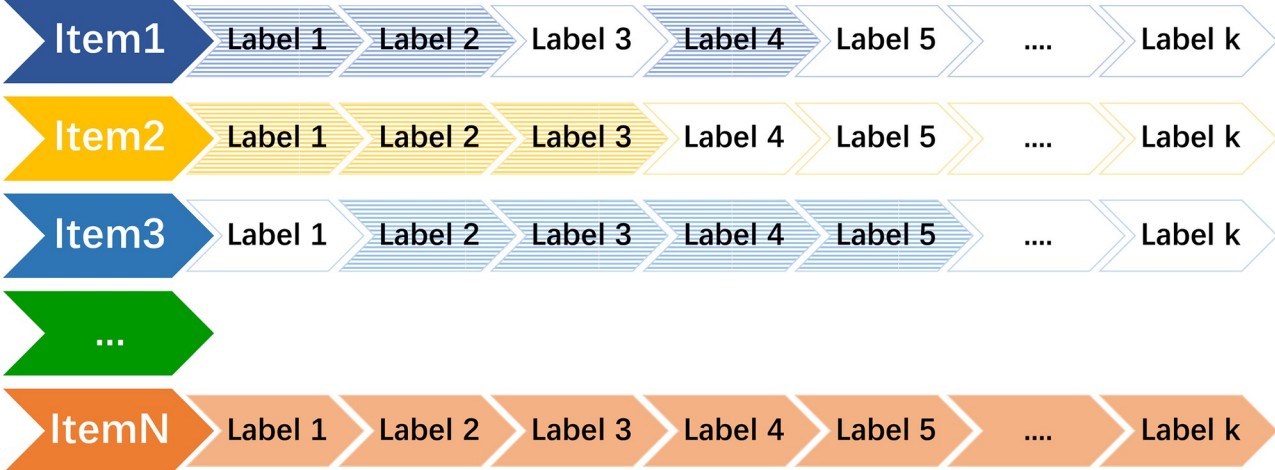

**Fig 3. Label vectors of items.** Item 1 includes labels 1, 2, 4, etc.; item 2 includes labels 1, 2, 3, etc.; and item 3 includes labels 2, 3, 4, 5, etc.

For each pair of users, we calculate a similarity vector based on their fuzzy preferences and the item's original label vector. Taking the user-based collaborative filtering algorithm as an example, for any two users $q$ and $w$, the similarity is a $k$-dimensional vector, defined as follows:

$$s_{qw} = \{s_1, s_2, \ldots, s_t, \ldots, s_k\} \tag{5}$$

where $s_1, s_2, \ldots, s_t, \ldots, s_k$ represent the similarity of user $q$, $w$ on label $l_1, l_2, \ldots, l_t, \ldots, l_k$ respectively. In order to facilitate the distinction, this paper refers to the user similarity vector $s_{qw}$ as the global similarity between users, and the user similarity $s_k$ on each label as the label similarity between users. Global similarity reflects the similarity of interests and preferences between users $q$, $w$, while label similarity reflects the similarity of users $q$, $w$ on a specific label $l_t$. Note that global similarity is a vector, while label similarity is a scalar.

In this method, the user's evaluation of the item is transformed into a pair of fuzzy preference indexes, deviating from the traditional subjective rating value. The global similarity between users is then calculated based on these two fuzzy preference indexes. To achieve this, we first calculate the similarity of the like index and dislike index between users using Eqs 6 and 7, respectively. Finally, we compute the average of the two similarities using Eq 9 to obtain the global similarity between users.

$$s_{qw}^{\text{like}} = T' \circ \sum_{i=1}^{n} [g_{qi}g_{wi}[1 - |\mu_1(r_{qi}) - \mu_1(r_{wi})|] \cdot L_i] \tag{6}$$

$$s_{qw}^{\text{dislike}} = T' \circ \sum_{i=1}^{n} [g_{qi}g_{wi}[1 - |\mu_2(r_{qi}) - \mu_2(r_{wi})|] \cdot L_i] \tag{7}$$

$$T' = \left\{ \frac{1}{\sum_{i=1}^{n} g_{qi}g_{wi} \cdot l_{i1}}, \frac{1}{\sum_{i=1}^{n} g_{qi}g_{wi} \cdot l_{i2}}, \cdots, \frac{1}{\sum_{i=1}^{n} g_{qi}g_{wi} \cdot l_{ik}} \right\} \tag{8}$$

$$s_{qw} = \frac{s_{qw}^{\text{like}} + s_{qw}^{\text{dislike}}}{2} \tag{9}$$

where $n$ indicates the number of items that users $q$ and $w$ have rated together, $g_{qi}$ and $g_{wi}$ indicate whether user $q$ and user $w$ rated item $i$, respectively. Let $g_{qi} = 1$ and $g_{wi} = 1$ if there is a score for item $i$ given by users $q$ and $w$, respectively; otherwise, $g_{qi} = 0$ and $g_{wi} = 0$. $L_i$ is the label vector of item $i$, and $l_{i1}$ is the first element in the label vector. $'\circ'$ represents the Hadamard product of the vectors. It is important to note that if there exists an item with a label that is not jointly rated by any users $q$ and $w$, the denominator of the element containing the corresponding label in the $T'$ vector is zero, which also makes the corresponding elements of the vector on the right side of the Hadamard product in Eqs 6 and 7 become zero. In such cases, we ignore the Hadamard product at this position, and the corresponding position of the vector on the left side in Eqs 6 and 7 is also set to zero. $\mu_1(r_{qi})$ and $\mu_2(r_{qi})$ represent user $q$'s like and dislike index of item $i$, respectively, which can be calculated using Eqs 6 and 7. Finally, we combine $s_{qw}^{\text{like}}$ and $s_{qw}^{\text{dislike}}$ using Eq 9 to obtain the global similarity between users $q$ and $w$.

Observing Eq 6, it becomes apparent that $|\mu_1(r_{qi}) - \mu_1(r_{wi})|$ only represents the absolute deviation of user $q$ and user $w$'s like index. When user $q$ and user $w$ have very similar preferences for item $i$, the absolute deviation becomes small, even becoming 0 if they have the same liking index for item $i$. Directly using this absolute deviation as the numerator in the equation would result in a proportionate impact, which is not conducive to similarity calculation. To address this, we introduce a standard offset value of 1 in the equation to amplify the deviation value of the two users. Hence, the denominator becomes $1 - |\mu_1(r_{qi}) - \mu_1(r_{wi})|$. This ensures a more logical similarity calculation, where smaller absolute deviation values of the like index between two users indicate greater similarity in their like preferences, leading to more accurate results. The same principle applies to Eq 7.

## Rating prediction based on similarity of fuzzy-preference label vectors

For rating predictions, we rely on the ratings of neighbor users for the target item. The neighbor users are defined as those who share the same rated items with the target user. Using the neighbor user's rating for the target item and their global similarity with the target user, we can predict the target user's rating for the target item.

In the prediction process, selecting an appropriate number of neighbors is crucial for the efficiency and speed of the recommender system. To explore the Fuzzy Preference Label Vector (FPLV) similarity algorithm, we compare three neighbor selection strategies. The first strategy, FPLV-ALL, involves using all neighbors of the target user for prediction. For scalability testing, we use the top K neighbors with the highest similarity (K = 100) in FPLV-KNN. If the neighbors are fewer than 100, all available neighbors are used for prediction. To assess accuracy, we apply a filter that keeps only neighbors with a Pearson similarity greater than 0, known as FPLV-SS method.

Different target items for the same user may contain different labels, resulting in varying fuzzy preferences from neighbors for these items and different label similarities. Thus, when predicting ratings, the similarity of the target item between the user and its neighbors varies based on the item's label. To simplify calculation, the vector global similarity $s_{qw}$ is converted into a scalar using Eq 10, and then the score is predicted using Eq 11.

$$s_{qwi} = \sqrt{\sum_{t=1}^{k} (s_{qwt})^2} \tag{10}$$

$$\check{r}_{qi} = \bar{r}_q + \frac{\sum_{w=1}^{N} \left[ (r_{wi} - \bar{r}_w) \cdot s_{qwi} \right]}{\sum_{w=1}^{N} s_{qwi}} \tag{11}$$

where $s_{qwt}$ represents the label similarity of users $q$ and $w$ on label $l_t$. If item $i$ does not contain label $l_t$, then $s_{qwt} = 0$. The calculation result $s_{qwi}$ in Eq 10 represents the influence of each neighbor of the target user on the final prediction, acting as the weight in the prediction value. In Eq 11, $\bar{r}_q$ and $\bar{r}_w$ are the average ratings of users $q$ and $w$, respectively, and $N$ is the total number of neighbors obtained for the target user $q$. Since the algorithm has a maximum of 100 neighbors, the numerator of Eq 11 can be interpreted as the product of the score deviation value of each neighbor and its influence on the target item.

## Rating prediction based on community information in similarity network

The previous article employed the fuzzy preference label vector similarity for prediction to enhance recommendation accuracy. However, using traditional similarity algorithms to select neighbors often leads to lower diversity in neighbor groups with higher similarity, while selecting neighbors with low similarity may sacrifice accuracy to increase diversity. To address this, complex network methods are used to potentially enhance diversity. As this study focuses on the user-based collaborative filtering similarity algorithm, a similarity network with users as nodes and similarity as edges is established. To improve prediction result diversity, understanding each user's distribution under diversity is essential.

After establishing the user-user similarity network, this paper uses the k-core decomposition method to determine the $k$ value for each node. The k-core algorithm simplifies complex networks and extracts highly correlated substructures. The k-core of a graph means that after removing nodes with a degree less than $k$, the remaining subgraph has all nodes with degree $k$. The degree in this context represents all users related to a certain user. Thus, a new user classification with $k$ values is obtained, ensuring high user diversity within the community.

We divide all users into communities using the k-core decomposition results [40]. Starting from the user with the lowest $k$ value, we find users with the same or higher $k$ value and select the one with the highest similarity. These users are then grouped into the same community. By traversing all users from low to high $k$ value, multiple communities composed of users with different $k$ values but the highest similarity can be obtained. The user similarity network example with 300 users is shown in Fig 4, where Fig 4(a) is the network before community division, and Fig 4(b) is the network after division. The node size in Fig 4(b) represents the $k$ value, and the color represents the community. The network is finally divided into 27 communities, improving prediction accuracy and enhancing recommendation diversity.

To improve the accuracy and diversity of recommendations, we utilize the known community information to weight the similarity results. The FPLV-SS neighbors are chosen as the basis for score prediction, and the similarity weight is incorporated into the prediction formula, resulting in the method called FPLV-C.

$$\check{r}c_{qi} = \bar{r}_q + \frac{\sum_{w=1}^{N}\left[(r_{wi} - \bar{r}_w)\cdot s_{qwi}\cdot \lambda\right]}{\sum_{w=1}^{N} s_{qwi}} \tag{12}$$

Each variable in the equation is consistent with Eq 11, where $\lambda$ denotes the weight of user similarity. In this paper, $\lambda$ is set to 1 if two users belong to the same community, and 0.7 otherwise, to strengthen the influence among users in the same community and maintain accuracy.

The specific algorithm process proposed in this paper is outlined in Algorithm 1.

**Algorithm 1** Link prediction algorithm based on similarity of fuzzy preference label vector and community information.

```
Input: User pair set U, rating set R, user item pair set Q, predicted
target user item pair P
Output: User q's predicted rating řqi for the target item i
```

(a)                                              (b)

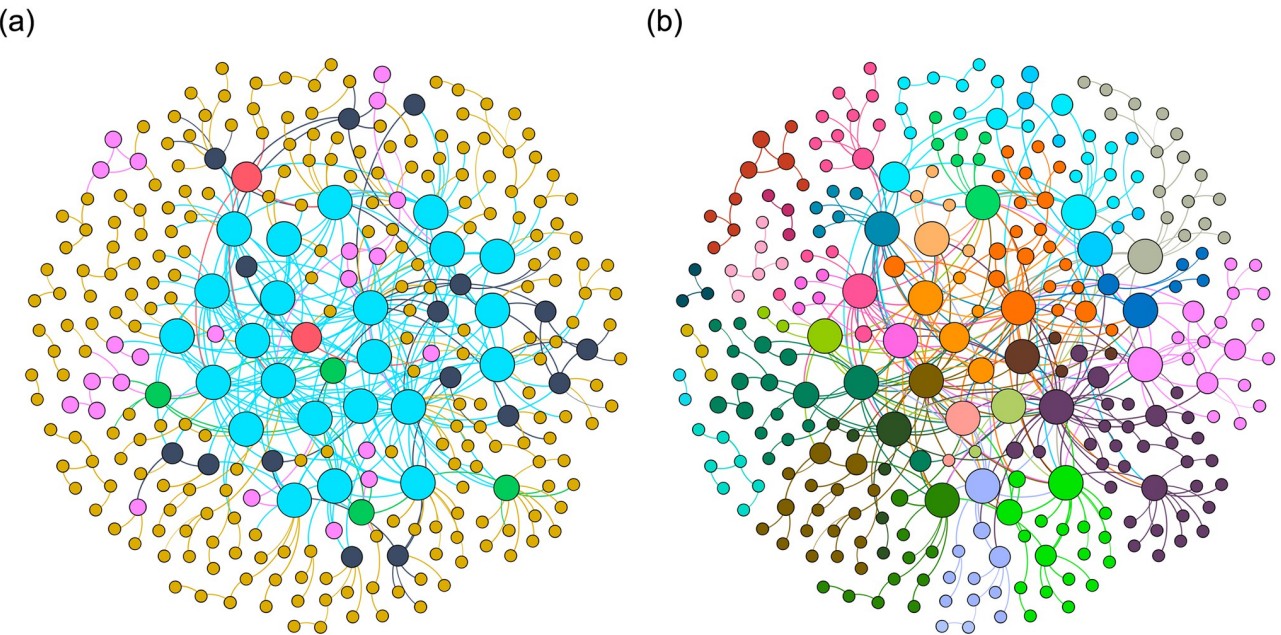

**Fig 4. User similarity network before and after partitioning communities.** Node size and color in Fig 4(a) represent 6 categories based on node $k$ values. In Fig 4(b), node size is categorized based on the $k$ value, and node color is categorized into 27 communities. Larger nodes indicate higher k values in both figures. (a) User similarity network and (b) User similarity network with community information.

```
 1: for each user item pair (q, i) ∈ Q do
 2:    r̄_q ← the average rating of user q
 3:    r_q^max ← the maximum rating of user q
 4:    r_q^min ← the minimum rating of user q
 5: end for
 6: for each user item pair (q, i) ∈ Q do
 7:    μ_1(r_qi) ← Eq 3
 8:    μ_2(r_qi) ← Eq 4
 9: end for
10: for each item i ∈ I do
11:    generate the original label vector T_i
12: end for
13: for each user pair (q, w) ∈ U do
14:    I_qw ← jointly rated item set
15:    Calculate the Pearson similarity PCC_qw
16:    for each item i ∈ I_qw do
17:       s_qw^like ← Eq 6
18:       s_qw^dislike ← Eq 7
19:       s_qw ← Eq 9
20:       s_qwi ← Eq 10
21:    end for
22: end for
23: Taking users as nodes and the similarity between users as edges,
    establish a user-user similarity network G
24: for each node n ∈ G do
25:    w_n ← community Id
26: end for
27: for each predicted target user item pair p_qi ∈ P do
```

```
28:    Take all users with PCC_qw > 0 as neighbor set V
29:    for each neighbor v ∈ V do
30:      if w_q ≠ w_v then
31:        λ = 1
32:      else
33:        λ = 0.7
34:      end if
35:    end for
36:    řc_qi ← Eq 12
37: end for
38: return User q's predicted rating řc_qi for the target item i
```

## Experiment setup

We implemented all the algorithms in this paper using the FSharp programming language with consistent caching technology and calculation processes. During the experiments, the target data was read into memory by the FSharp code, randomly screened, and then divided for five-fold cross-validation. The whole data set is randomly divided into five groups for experiments, any one of which is used as the test set for the remaining four groups, and the average value of the five experiments is used as the final result of the experiment. Each data division underwent preheating using the average score for prediction before executing the learning and test procedures for each predictive algorithm sequentially, ensuring a standardized interface to minimize code implementation impact on performance.

After each test, various parameters were tested for prediction and recommendation, and the results were recorded on the hard disk. The recording process was excluded from time-cost statistics. Subsequently, the algorithm prediction results and cache were cleared, and the next algorithm prediction and verification process was carried out one by one until all algorithms completed. Data analysis and visualization were performed using a Python program to present the final comparison of experimental results.

The experiment was conducted on a computer equipped with an AMD 5950X CPU with 16 cores and 128GB of memory. All algorithms utilized the same parallel computing structure, enabling parallel computation in both user similarity calculations and multiple user-item target predictions.

### Datasets

This paper uses two well-known datasets to evaluate and verify the experimental results, which are all from the real world and have been widely used in other prediction algorithms research.

1. MovieLens-25M [50]: The dataset collected and maintained by GroupLens Research. It contains 25 million movie ratings from 162,000 users on 62,000 movies. The minimum score is 0.5 points, the maximum is 5 points, and the increment is 0.5 points. The data sparsity amounts to approximately 99.75%, and the complete dataset is available for download at https://grouplens.org/datasets/movielens/25m/.

2. Netflix Dataset [51]: It contains 100480507 ratings of 17770 movies by 480189 users. The minimum user rating is 1 point, the maximum is 5 points, and the increment is 1 point, provided by the Netflix Prize recommendation algorithm competition. The data sparsity is about 98.82% and the dataset is available for download at https://www.kaggle.com/netflix-inc/netflix-prize-data.

To expedite the experiment, a random sample of 5000 users and their corresponding ratings was selected from the dataset. For CF algorithms, the experiments involve selecting $k$ nearest

neighbors for prediction and recommendation, with $k$ ranging from 1 to 200. However, for algorithms that use all qualified neighbors, their performance remains unchanged regardless of the number of selected neighbors.

## Comparative baseline algorithms

**Similarity-network resource allocation.**   By forming a similarity network based on user-item relationships, SRA algorithm [45] calculates degree centrality and community structure to identify accurate neighbors efficiently. The degree centrality is balanced using resource allocation, and prediction results are corrected using normalized degree values and communities.

SRA similarity in Eq 13 consists of coefficients and classic PCC Pearson similarity coefficients.

$$SRA(u, v) = 2.0 \cdot \frac{RA_{ui}(u, v) \cdot RA_{uv}(u, v)}{RA_{ui}(u, v) + RA_{uv}(u, v)} \cdot PCC(u, v). \tag{13}$$

where $RA_{uv}(u, v)$ and $RA_{ui}(u, v)$ are two coefficients given by Eqs 14 and 15.

$$RA_{uv}(u, v) = 1.0 + \sum_{n \in N_u \cap N_v} \frac{1}{k_n}, \tag{14}$$

where $N_u$ denotes the set of nodes (neighbors) of user $u$ that have connected edges in the similarity network, $n \in N_u \cap N_v$ denotes the set of common neighbors of user $u$ and user $v$, and $RA_{uv}(u, v)$ denotes their neighbor reliability coefficient.

$$RA_{ui}(u, v) = \sum_{i \in I_u \cap I_v} \frac{1}{d_i}, \tag{15}$$

where $I_u$ represents the set of items chosen by user $u$, $I_v$ represents the set of items chosen by user $v$, and $d_i$ represents the degree value of item $i$.

**Vector similarity collaborative filtering.**   Vector similarity [46] introduces a novel way to calculate user similarities using a vector-based measurement, considering multiple dimensions based on item attributes. Global similarity, local similarity, and meta similarity are defined to create a vector indicator of user similarity, with the distinguishing feature that similarity varies with different target items.

$$S_{ij}^{\alpha} = S_{ij} \circ T_{\alpha}. \tag{16}$$

where $S_{ij}^{\alpha}$ is the local similarity between $u_i$ and $u_j$ for the prediction of $u_i$'s rating on $i_{\alpha}$. $S_{ij}$ is the global similarity between $u_i$ and $u_j$. And the $'\circ'$ stands for the Hadamard product of the vectors. It can be found that the local similarity between users tends to be different due to the difference of tag vectors $T_{\alpha}$ of the item.

$$S_{ij} = g\left( \sum_{\alpha=1}^{n} \beta_{i\alpha} \beta_{j\alpha} \right) \circ \left[ \frac{\sum_{\alpha=1}^{n} \beta_{i\alpha} \beta_{j\alpha} [10 - |(r_{i\alpha} - \bar{r}_i) - (r_{j\alpha} - \bar{r}_j)|] T_{\alpha}^T}{\sum_{\alpha=1}^{n} \beta i\alpha \beta_{j\alpha} T_{\alpha}^T} \circ TC. \right] \tag{17}$$

where $TC = \phi(\sum_{\alpha=1}^{n} \beta_{i\alpha} \beta_{j\alpha} T_{\alpha}^T)$, $S_{ij}$ is the similarity between $u_i$ and $u_j$, and $g(n_{ij}) = ln^{n_{ij}}$.

**Entropy collaborative filtering.**   The information entropy of user ratings reflects the overall rating behavior of users on items. Soojung Lee combined the information entropy of user ratings with recommendation algorithms in his research [52], and improved it based on Pearson similarity. The study believes that when calculating the similarity between two users, if the item entropy value they jointly rated is higher, their similarity will be greater, otherwise, the

similarity result will be smaller. Similarly, when the entropy of an item is small, the greater the difference in ratings, the less similarity between users. The information entropy of item ratings can be calculated by the Eq 18 and the Eq 19.

$$E(i) = -\sum_{r=r_{min}}^{r_{max}} prob(r_i = r) \cdot \log_2(prob(r_i = r)) \tag{18}$$

$$prob(r_i = r) = \frac{|\{u \in U \mid r_{ui} = r\}|}{|\{u \in U \mid r_{ui} \in [r_{min}, r_{max}]\}|} \tag{19}$$

where $r = r_{min}$ and $r_{max}$ represent the maximum and minimum user ratings respectively, $U$ represents the set of all users, $prob(r_i = r)$ represents the likelihood that the user will rate the item $i$ as.

The improved similarity calculation method is shown in the Eq 20.

$$COR_{ent}(q, w) = \frac{\sum_{i \in I_{qw}} (r_{qi} - \bar{r}_q) \cdot (r_{wi} - \bar{r}_w) \cdot E(i)}{\sqrt{\sum_{i \in I_{qw}} (r_{qi} - \bar{r}_q)^2} \sqrt{\sum_{i \in I_{qw}} (r_{wi} - \bar{r}_w)^2}} \tag{20}$$

where $r_{qi}$ and $r_{wi}$ represent the ratings of user $q$ and user $w$ on item $i$, respectively, $\bar{r}_q$ and $\bar{r}_w$ represent the average ratings of users $q$ and $w$, respectively. $I_{qw}$ represents the set of items jointly rated by users $q$, $w$, and $E(i)$ represents the information entropy of item $i$.

**DTEC-SCoR algorithm.** Dual Training Error Correction (DTEC) takes into account the error between users and items in the training set [53]. DTEC computes a model that makes the recommendation error in the training set zero, and then applies it to the Synthetic Coordinate recommender system (SCoR) to improve rating predictions. This approach is applicable to any model-based recommender system with positive training error. The DTEC method can perform error correction from both user and item perspectives, thus proposing a dual system that effectively combines the two corrections, and its prediction method is shown in the Eq 21.

$$r'(q, i) = \hat{r}(q, i) + \frac{c_i(q) + c_q(i)}{2} \tag{21}$$

where $r'(q, i)$ is the predicted rating of the item $i$ by the user $q$, and $\hat{r}(q, i)$ represents the predicted rating of the initial recommender system. $c_i(q)$ and $c_q(i)$ represent user-based training error correction (UTEC) and item-based training error correction (ITEC), respectively. The calculation methods are shown in the Eq 22 and the Eq 23 respectively.

$$c_q(i) = \sum_{k=1}^{|TR_q|} w(i, k) \cdot [r_q(i_k) - \hat{r}(q, i_k)] \tag{22}$$

$$c_i(q) = \sum_{k=1}^{|TR_i|} w(q, k) \cdot [r_i(q_k) - \hat{r}(q_k, i)] \tag{23}$$

where $r_q(i_k)$ and $r_i(q_k)$ represent the user $q$'s rating on the item $i_k$ and the user $q_k$'s rating for item $i$ in the training set. $|TR_q|$ and $|TR_i|$ represent the number of items rated by user $q$ and the number of times the item $i$ was rated in the training set, respectively. $w(i, k)$ and $w(q, k)$ are weight coefficients for normalization, which can be understood as the impact of item $i_k$ on UTEC and user $q_k$'s impact on ITEC.

## Performance evaluation metrics

The recommender system output contains extensive information, necessitating evaluation from multiple perspectives to assess its effectiveness. This paper introduces six metrics to measure prediction accuracy, sorting accuracy, and diversity: MAE, RMSE, F1, Half Life Utility (HLU), Sorting Accuracy (SA), and Degree Diversity.

**Prediction accuracy.** MAE and RMSE are widely used evaluation methods in collaborative filtering recommender systems. MAE measures the average absolute error between the predicted score and the actual score given by users. The overall system performance is reflected by the average MAE of all users. RMSE, on the other hand, emphasizes the impact of larger error values on prediction accuracy. The calculation methods for MAE and RMSE are shown in Eqs 24 and 25 respectively.

$$MAE = \frac{1}{n}\sum_{i=1}^{n}|r_{qi} - \check{r}_{qi}| \tag{24}$$

$$RMSE = \sqrt{\frac{\sum_{i=1}^{n}(\check{r}_{ui} - r_{ui})^2}{n}} \tag{25}$$

where $n$ is the number of samples in the test set, $r_{qi}$ represents the user $q$'s predicted score for the item $i$ in the test set, and $\check{r}_{qi}$ represents the actual rating of user $q$ for item $i$ in the test set. The smaller the results of MAE and RMSE, the higher the prediction accuracy and the better the algorithm.

F1-score, Precision Rate, and Recall Rate are commonly used in algorithm evaluation. Users' ratings for items are classified as liked, disliked, or unrated. Typically, unrated items are grouped with disliked items, treating them as non-relevant items. To calculate these metrics, items with user ratings higher than their average rating are considered liked or relevant items. This approach is effective for binary or univariate ratings.

The evaluation method categorizes items into four groups, as shown in Table 1. For relevant items, if the system recommends the item, it is marked as True Positive (TP), otherwise as False Negative (FN). For irrelevant items (i.e., items the user does not like and should not be recommended), if the system makes a recommendation, it is marked as False Positive (FP), otherwise as True Negative (TN).

The Precision Rate measures the ratio of relevant items among all recommended items, while the Recall Rate calculates the ratio of all relevant items that the system recommends. The calculation formulas for Precision Rate and Recall Rate are shown in Eqs 26 and 27, respectively.

$$P = \frac{TP}{TP + FP} \tag{26}$$

**Table 1. For relevant and irrelevant items, classify items into four categories based on whether the system makes a recommendation.**

|  | Recommended | Not recommended |
|---|---|---|
| Related(User likes) | TP | FN |
| Not related(User don't like) | FP | TN |

$$R = \frac{TP}{TP + FN} \tag{27}$$

To address the limitation of precision and recall metrics requiring users to rate all items, Sarwar et al. proposed an enhanced evaluation method [54] for recommender systems. The calculation formulas are shown in Eqs 28 and 29.

$$P@N_q = \frac{TP}{N} \tag{28}$$

$$R@N_q = \frac{TP}{|R_e l_q|} \tag{29}$$

where $N$ represents the number of items in the recommendation list, and $|R_e l_q|$ represents the set of related items of the user $q$, that is, the items that the user $q$ likes. In the classification accuracy evaluation method, an item that is not present in the user's true rating may not necessarily be irrelevant, as the user might not have noticed the item rather than disliking it. To balance precision and recall, research often employs $F1_q$ [54]. $F1_q$ combines $P@N_q$ and $R@N_q$, and its calculation method is shown in Eq 30. A larger $F1_q$ result indicates better algorithm performance.

$$F1_q = \frac{2(P@N_q) \cdot (R@N_q)}{(P@N_q) + (R@N_q)} \tag{30}$$

**Ranking quality of recommendation list.**   Half Life Utility (HLU) refers to the time it takes for the concentration of a specific substance to reduce to half of its initial value after a certain reaction. In the context of recommendation lists, higher-ranked items are more likely to be noticed by the user, causing the probability of lower-ranked items being viewed to drop rapidly. This metric considers the exponential relationship between the probability of a user browsing a product and its position in the recommendation list, allowing it to measure the extent of difference between the user's actual rating and the system's predicted rating. The HLU calculation method for the target user is given by Eq 31.

$$HLU_q = \sum_{i=1}^{N} \frac{max(r_{qi} - \bar{r}_q, 0)}{2^{\frac{i-1}{h-1}}} \tag{31}$$

where $h$ is the threshold of half-life, and $\bar{r}_q$ represents the average rating of users. To obtain the half-life result of the recommender system, calculate the half-life values for all users using Eq 32. A higher value indicates a better sorting ability of the system. HLU is a ranking-based evaluation metric in this study, where the threshold for half-life is set to $h = 2$ in the experiment.

$$HLU_{total} = \frac{\sum_{q=1}^{m} HLU_q}{m} \tag{32}$$

Sorting Accuracy (SA) focuses on the entire recommendation list's order. It considers an item's ranking position correct if its score is not lower than all items that appear after it in the list. SA measures the percentage of items in the correct position in the recommendation list,

with a higher value indicating better performance.

$$OS(S_i) = \prod_{j=i+1}^{|L|} I(S_i \geq S_{j+1}) \tag{33}$$

where $S_i$ and $S_j$ are the actual ratings of the items in the $i$ and $j$ positions by the user in the recommendation list. $I$ is the indicator function, where its value is 1 when the conditions in parentheses are met, and 0 otherwise. $L$ represents the recommendation list, and $OS(S_i)$ denotes the evaluation value of the item at position $i$ in the recommendation list.

Based on this assumption, a higher SA value indicates that the ranking of items in a recommendation list closely matches the user's ranking of those items. Therefore, a higher SA value signifies that the recommender system can provide more accurate item recommendations to users. The SA value reflects the overall accuracy of the recommender system's combined recommendations. By calculating the evaluation value for each item in the recommendation list, the SA value for that list can be obtained using Eq 34.

$$SA = \frac{\sum_{i=1}^{|L|} OS(S_i)}{|L|} \tag{34}$$

where the numerator can be interpreted as the total number of correctly sorted items in the recommendation list, and the denominator is the length of the current recommendation list, that is, the number of items in the list. It can be seen that the higher the SA value, the more accurate the system recommends the list.

**Recommendation diversity.** Diversity plays a crucial role in assessing recommender system performance, as excessive similarity among recommended items can lead to user monotony and negatively impact user experience. An effective recommender system should not only encompass the types of items that users prefer but also introduce less popular items to cater to users' diverse interests. To measure diversity, we utilize Item Degree Diversity (IDD) to evaluate item popularity and Item Genre Diversity (IGD) to assess item type coverage in the recommendation list.

IDD is an evaluation metric based on item degree value, which quantifies an item's popularity based on the number of times it has been rated. Evaluating the degree value diversity of the recommendation list involves calculating the difference between the degree values of each item. A greater degree value difference among items indicates a higher level of diversity in the recommendation list. The specific calculation method is provided in Eq 35.

$$IDD = \frac{\sqrt{\frac{1}{|L|-1} \sum_{k=1}^{|L|} \left(d_k - \bar{d}\right)^2}}{\bar{d}} \tag{35}$$

where $|L|$ represents the length of the recommended list, that is, the number of items in the list. $d_k$ represents the degree value of the $k$th item, and $\bar{d}$ represents the mean degree value of all items in the recommendation list. This equation calculates the overall standard deviation of the item degree values in the list, divided by the mean of the item degree values. This ratio reflects the dispersion of the item degree values relative to the unit mean in the recommendation list. A higher IDD value indicates a greater likelihood of recommending unpopular items and tapping into users' potential interests.

## Results and discussion

We evaluate the method proposed in this paper through various dimensions. The primary focus is on the accuracy of rating prediction and the ranking quality of items in the

recommendation list. Additionally, we analyze the diversity of the recommendation list and assess the algorithm's scalability by studying its time complexity and observing its performance in our experiments.

## Prediction accuracy comparison

Fig 5 presents a comparison of MAE results between the algorithm proposed in this paper and other recent algorithms on different datasets. Fig 5(a) displays the results obtained using the MovieLens-25M dataset, while Fig 5(b) shows the results from the Netflix dataset. Notably, the FPLV-KNN and Entropy algorithms achieve the lowest prediction errors in both datasets. The FPLV-KNN algorithm requires approximately 45 neighbors to achieve the lowest error, whereas the Entropy algorithm requires over 120 neighbors. The parameter-free FPLV-SS algorithm also outperforms other comparison algorithms in error control. On the other hand, FPLV-ALL, which considers all optional neighbors, exhibits a weaker suppression of prediction errors, while FPLV-C, which incorporates similarity network features, shows no advantage in error level and falls within a moderate range.

Likewise, Fig 6 illustrates the RMSE results of the proposed algorithm and other compared algorithms. RMSE, compared to MAE, emphasizes the impact of prediction scores with larger errors on the recommendation results. Fig 6(a) and 6(b) display the RMSE results for different algorithms on the two datasets. The FPLV-KNN algorithm shows the best error suppression. It performs slightly worse than Entropy on the MovieLens dataset, which has more dense ratings, but performs at a similar level on the sparser Netflix dataset. The performance of FPLV-C, which introduces similarity network information, is still inferior to that of FPLV-SS, suggesting that the inclusion of community information increases the likelihood of errors.

In a denser dataset like MovieLens, the errors of each algorithm tend to be similar. However, in a sparse dataset like Netflix, the difference in errors between each algorithm becomes more pronounced.

## Ranking quality comparison

Fig 7 illustrates the comparison of HLU results achieved by the proposed algorithm and other algorithms on various datasets. The algorithms designed in this paper, with the exception of FPLV-ALL, demonstrate excellent performance, comparable to the top-performing SRA algorithm. The introduction of the similarity network feature in FPLV-C is particularly rewarding, as it significantly improves the likelihood of user's favorite items being ranked at the top of the recommendation list. The experiment on another dataset, as shown in Fig 7(b), exhibits similar characteristics, with each algorithm performing consistently well across both datasets.

We utilize the SA parameter to evaluate the overall ranking correctness of the user recommendation list.

Fig 8(a) and 8(b) present a comparison of SA values obtained by the proposed FPLV series algorithms and other benchmark algorithms on different datasets. The results demonstrate that the FPLV algorithms outperform the other benchmark algorithms in terms of SA values.

The performance of FPLV-C and FPLV-SS is identical, while FPLV-KNN achieves slightly higher values. However, it is worth noting that the first two algorithms do not require any parameter tuning. Upon examining the definitions of HLU and SA, we observe that the introduction of similarity communities does not impact the overall sorting correctness. Nevertheless, it enables users to find their favorite items among the top recommendations.

In both data sets, the VS algorithm exhibits a decrease in sorting accuracy as the number of neighbors increases, indicating that the VS algorithm struggles to accurately measure less similar neighbors.

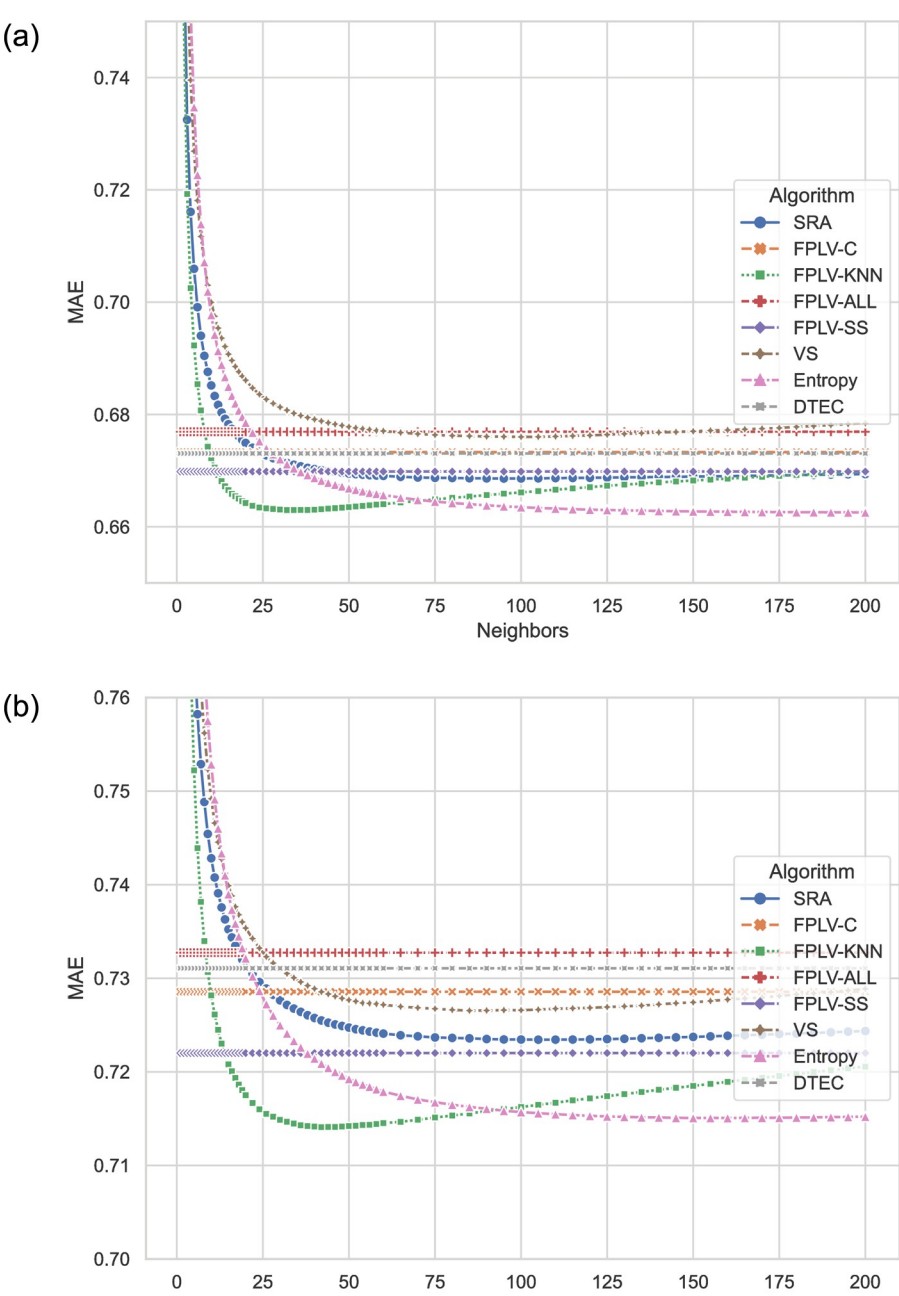

**Fig 5. Comparison of MAE results for different algorithms with varying numbers of neighbors.** (a) MovieLens and (b) Netflix.

## Classification accuracy comparison

Fig 9(a) and 9(b) are F1 results obtained by the algorithm proposed in this paper and other algorithms in recent years under different data sets.

In this experiment, the VS and DTEC algorithms demonstrate the best performance in terms of F1, which comprises Precision and Recall. Upon closer examination, it is evident that the Recall values of both VS and DTEC algorithms are significantly superior to those of other algorithms, while the Precision values among all algorithms show minor differences. As a

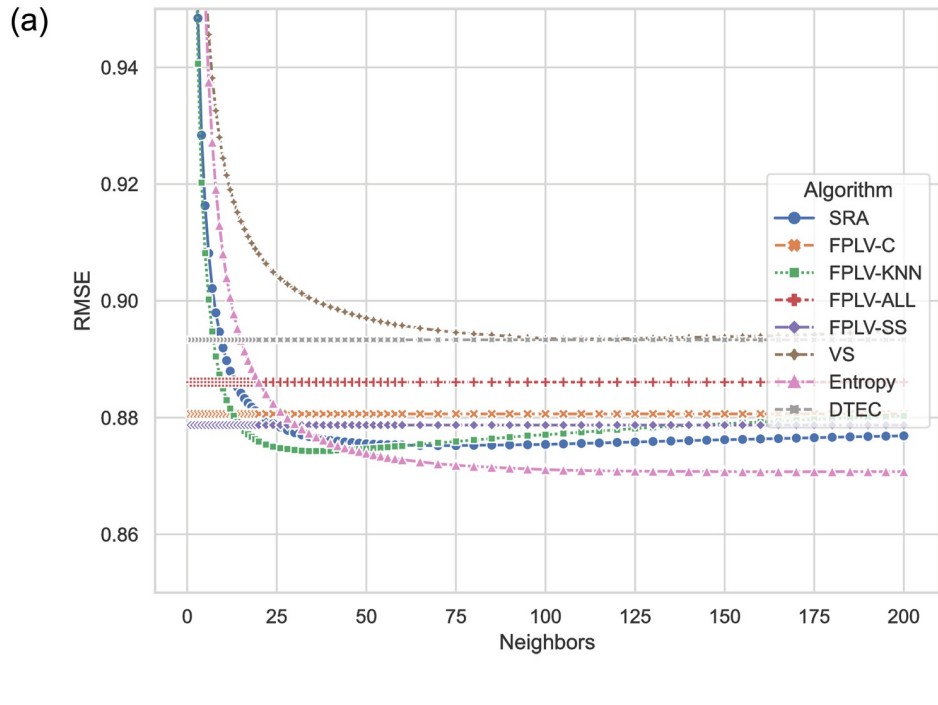

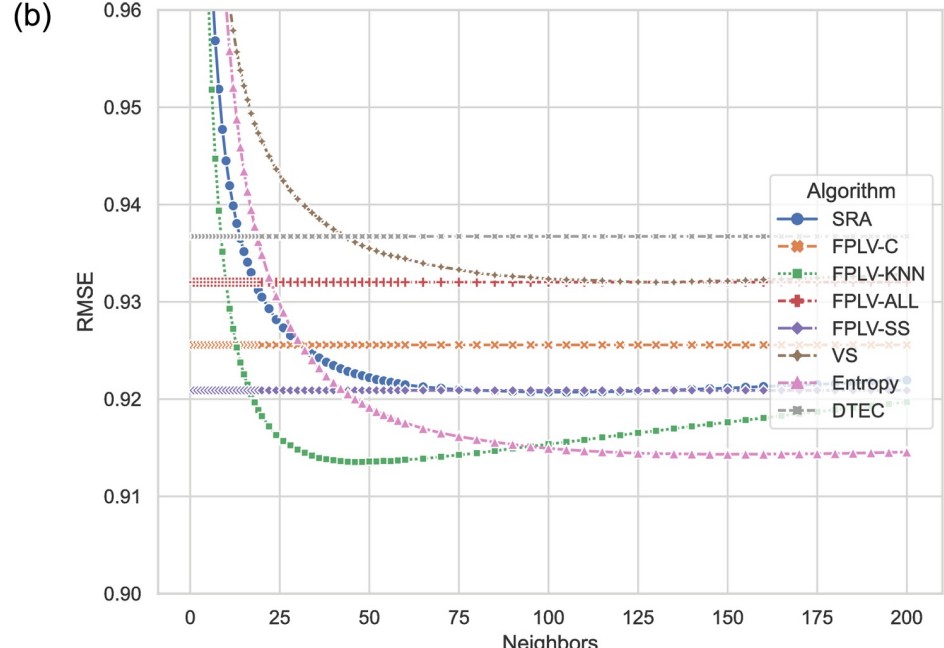

**Fig 6. Comparison of RMSE results for different algorithms with varying numbers of neighbors.** (a) MovieLens and (b) Netflix.

result, the advantages of VS and DTEC algorithms in F1 can be attributed to their ability to minimize the likelihood of missing items that users prefer.

The FPLV-KNN algorithm exhibits excellent performance and stands out in the top tier of algorithms. However, several other FPLV algorithms show average performance, with no significant advantage in classification accuracy. The main contributing factor is their relatively

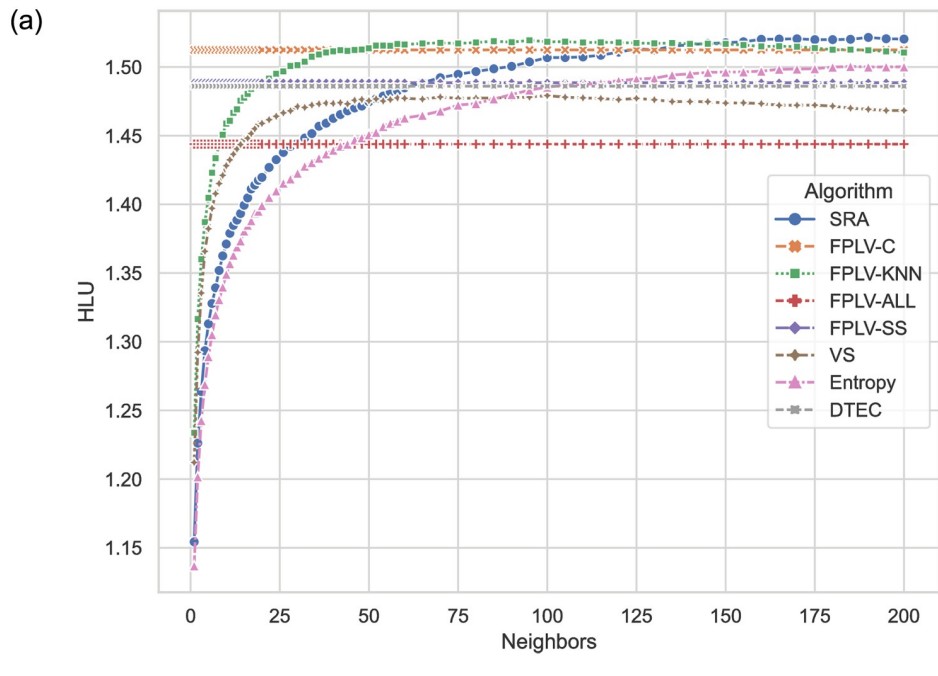

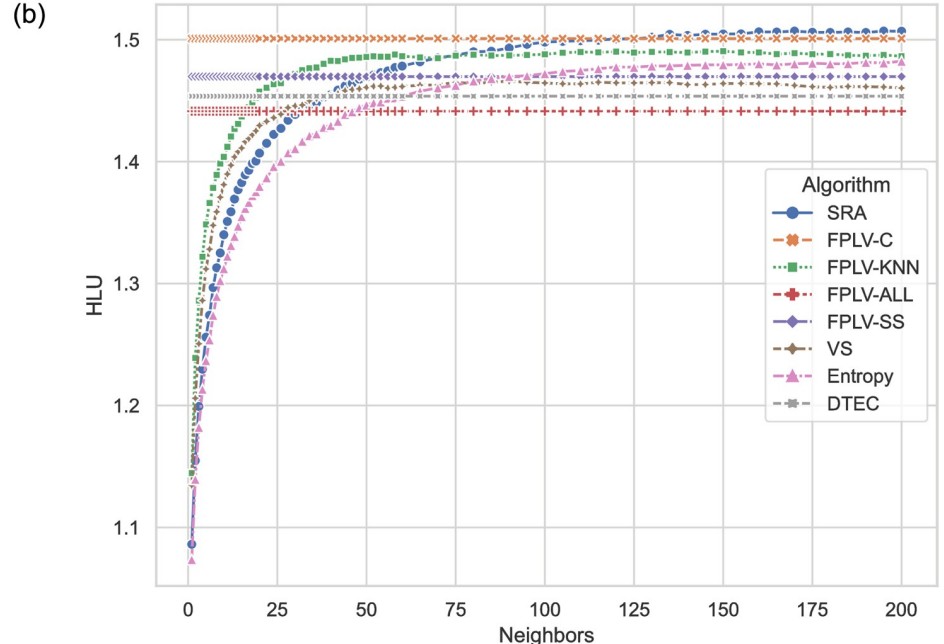

**Fig 7. HLU results of different algorithms on various datasets.** (a) MovieLens and (b) Netflix.

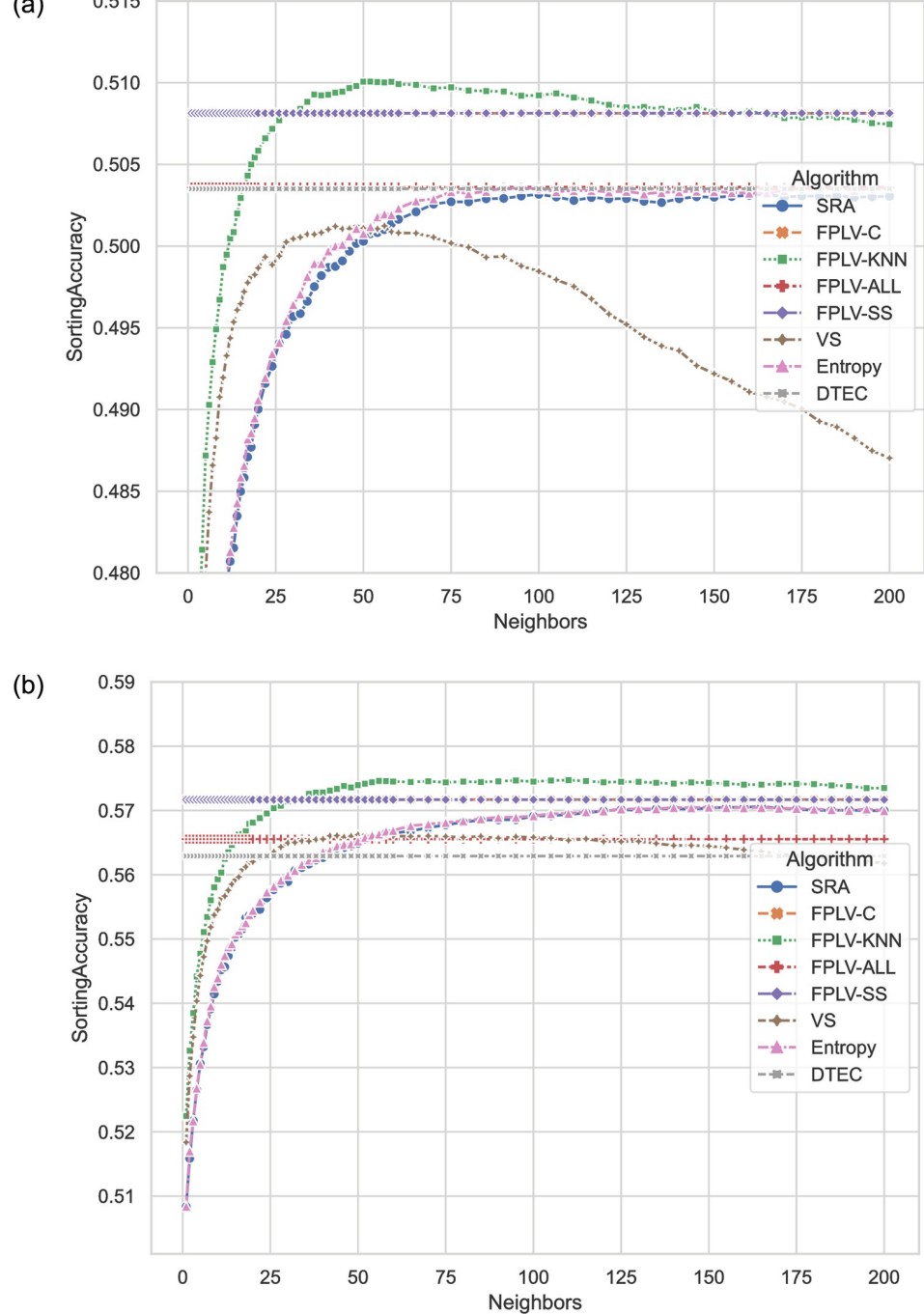

**Fig 8. SA values of different algorithms when selecting different numbers of neighbors.** (a) MovieLens and (b) Netflix.

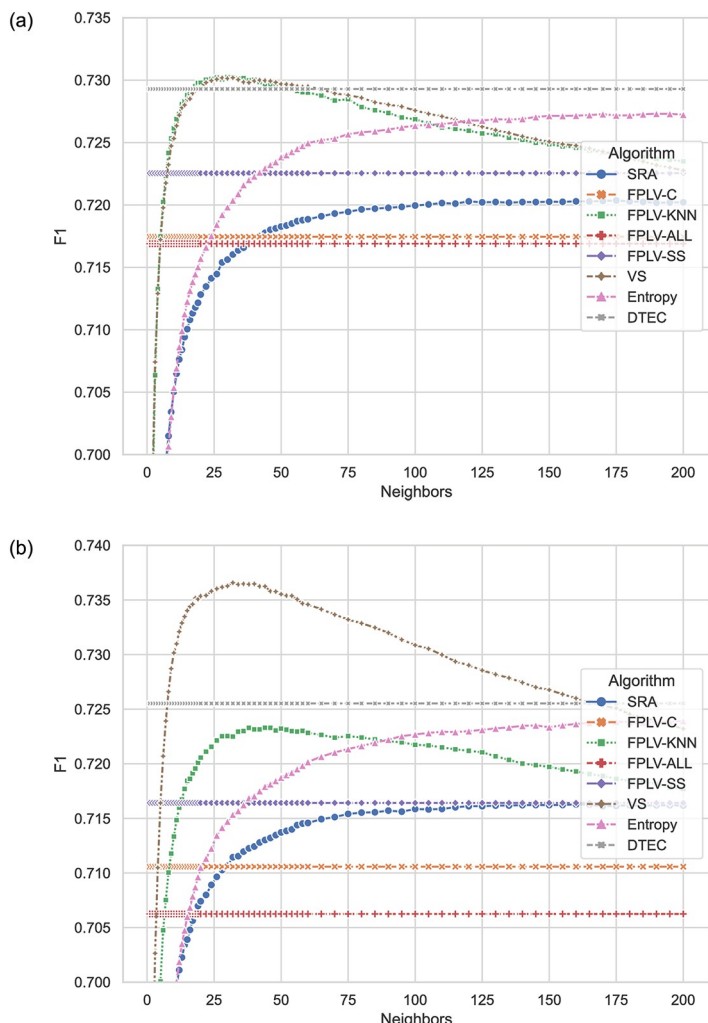

**Fig 9. Comparison of F1 scores for different algorithms with varying numbers of neighbors.** (a) MovieLens and (b) Netflix.

low recall scores. These algorithms tend to underestimate unknown ratings, leading to the omission of items that some users would actually prefer.

## Diversity comparison

Fig 10(a) and 10(b) show a comparison of item degree diversity results obtained by the algorithm proposed in this paper and other algorithms in recent years using different datasets. The results indicate that FPLV-C outperforms other FPLV algorithms in terms of item degree diversity, demonstrating that the introduction of community characteristics can enhance the diversity of user lists.

## Evaluating algorithm scalability

The scalability of an algorithm is determined by its learning time and data storage requirements during prediction. In this experiment, we assess the scalability of each algorithm by

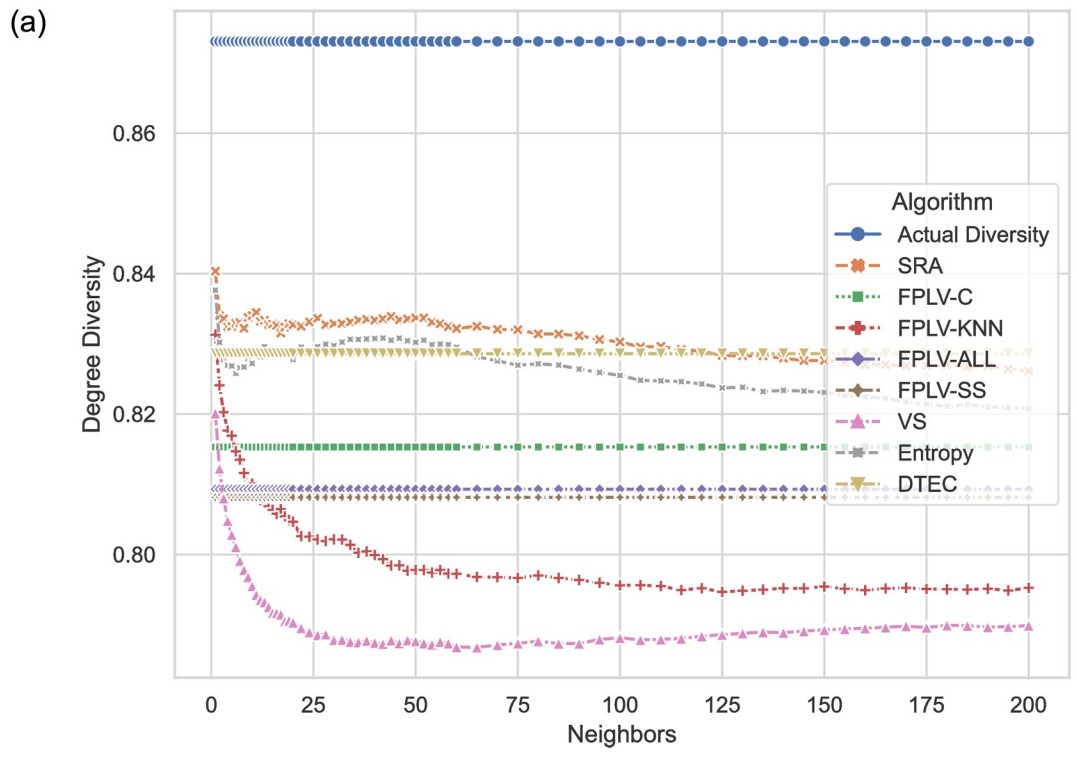

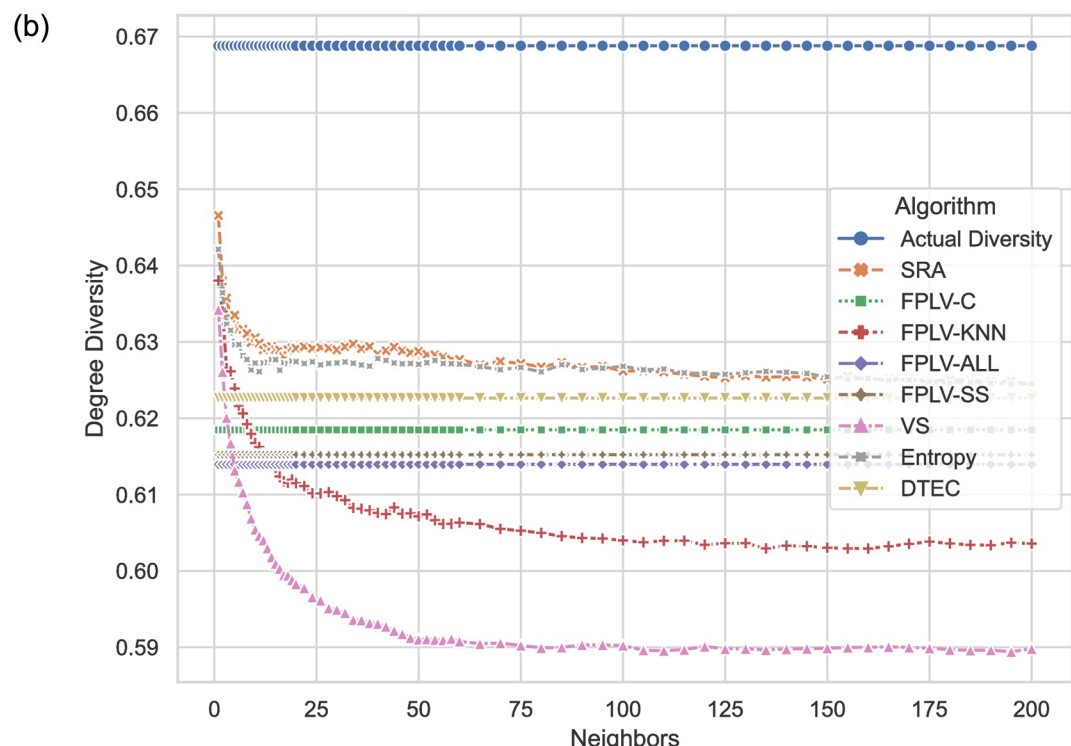

**Fig 10. Variation in Item Degree Diversity (IDD) for different algorithms and neighbor selections.** (a) MovieLens and (b) Netflix.

analyzing their time complexity and measuring the actual time consumed during the experiment. A more efficient algorithm with lower learning time and reduced data storage needs demonstrates better scalability.

The time complexity analysis of the algorithms used in this study reveals the following insights: FPLV-KNN and FPLV-ALL have a time complexity of $O(m) + O(n^2)$, where $m$ represents the number of scores and $n$ is the number of users. SRA has a time complexity of $2O(m) + O(n^2)$ due to the calculation of user similarity networks. VS requires $2O(n^2)$ for vector similarity calculations. Entropy has a time complexity of $O(nm)$ due to additional entropy value calculations. FPLV-SS and FPLV-C have the highest time complexity, $O(m) + 2O(n^2)$, due to the computation of Pearson similarity. Finally, DETC has a time complexity of $p \cdot O(m^2)$, which is influenced by the number of scores. Sparse scores result in faster computation speed.

Fig 11(a) and 11(b) display the real-world time spent in our experiments, revealing that FPLV-SS and FPLC-C exhibit similar time complexity, and their computation does not decrease when the ratings are sparse. Their time complexity is on par with other methods. Moreover, FPLV-KNN demonstrates the lowest time complexity and performs exceptionally well in most of our experiments. The only drawback of FPLV-KNN is that the number of neighbors needs to be determined through experiments.

## Conclusion

In this paper, we introduce a novel approach leveraging user fuzzy preferences and item label vectors to enhance predictive accuracy in recommender systems, particularly in addressing the challenge of ranking performance in recommendation lists. By combining fuzzy preferences with item label vectors, we construct user-to-item fuzzy preference label vectors, effectively capturing users' uncertain and imprecise preferences. Additionally, we design a range of similarity measures for fuzzy preference label vectors, including parameter-free methods, which significantly improve the accuracy of similarity calculations in recommender systems. Leveraging the user-user similarity network, our method effectively balances prediction accuracy and recommendation diversity by utilizing various information from the recommender systems.

Our algorithm excels in ranking performance of recommendation lists and demonstrates superior rating prediction accuracy compared to classical heuristic methods. Furthermore, the FPLV series algorithms maintain commendable performance in the classification accuracy metric F1 while achieving a moderate level of diversity in the recommendation lists.

However, our approach requires the availability of item labels for making accurate predictions and recommendations. Additionally, the parameter-free methods' performance in the classification prediction index F1 is not as strong as expected, mainly due to better precision but poorer recall compared to the VS and DTEC algorithms. The reasons behind this observation remain inconclusive in the current study and warrant further investigation in future research. Nonetheless, our proposed algorithm represents a significant step forward in improving the effectiveness and efficiency of recommender systems.

In future research, we identify several areas that warrant further investigation:

1. Expanding Fuzzy Preferences: Currently, our fuzzy preference approach only considers binary preferences (like and dislike). To enhance accuracy, we plan to explore multi-category fuzzy preferences, incorporating additional transition preferences between like and dislike. This will offer a more nuanced understanding of user preferences, reducing subjectivity and improving similarity calculation accuracy.

2. Weighted Label Attributes: The label vector of items introduced in our work expands the role of scoring. To better represent item characteristics, we intend to introduce weights to labels, enabling more accurate rating predictions based on item features.

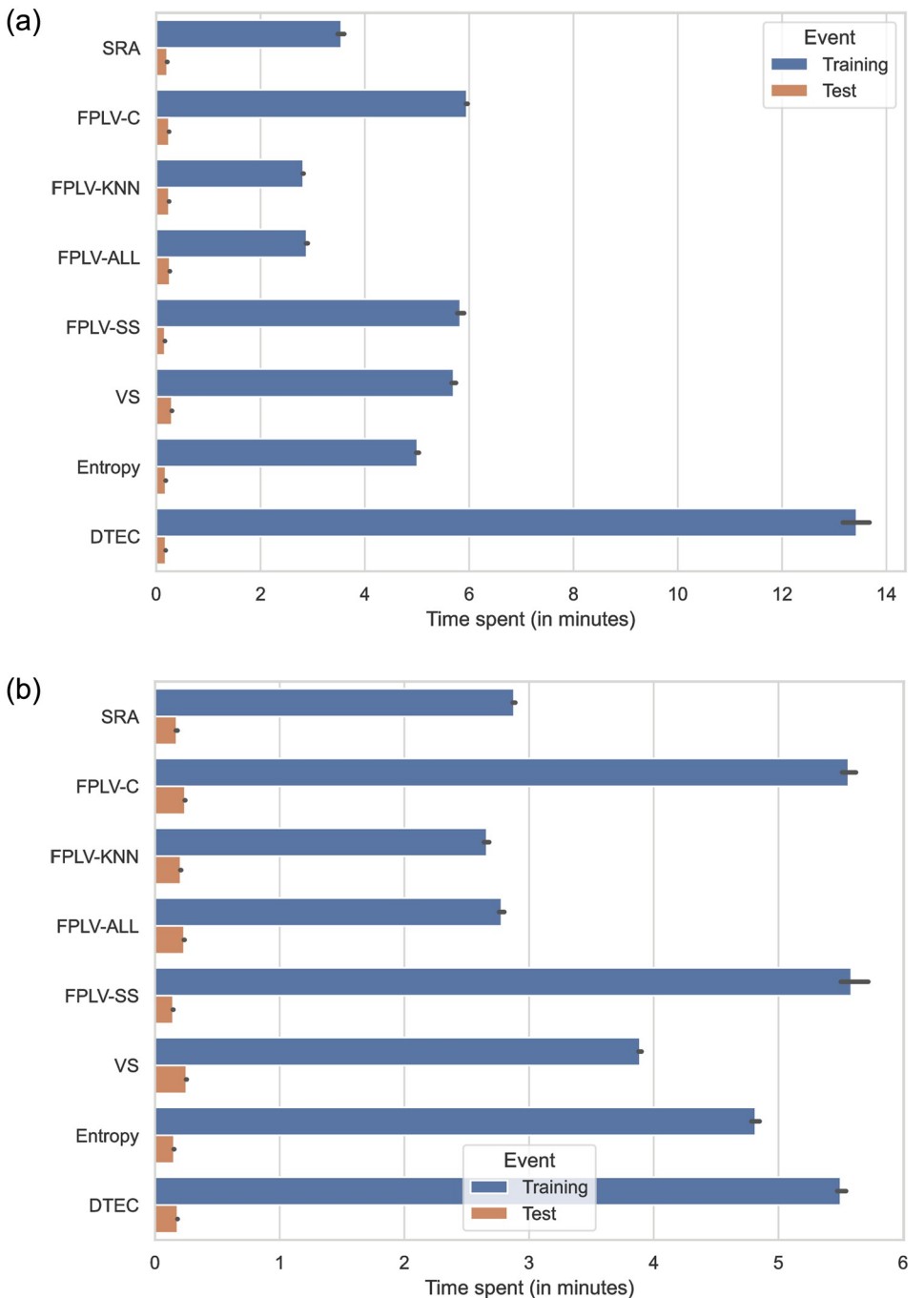

**Fig 11. Comparison of prediction time for different algorithms on MovieLens-25M and Netflix datasets (unit: Minutes).** The black lines represent the upper and lower bounds of the distribution obtained from multiple trials in the five-fold cross validation. (a) MovieLens and (b) Netflix.

3. Expanding Complex Network Theory: While the use of complex networks has already improved the performance of our recommender system, we aim to explore the integration of complex network theory on a larger scale to unlock new possibilities in recommender systems.

Acknowledging the limitations in time and resources, we recognize that certain aspects may not be fully perfected. Therefore, these proposed areas for further research will address these limitations and contribute to enhancing the effectiveness and accuracy of the recommender system.

## Acknowledgments

We extend our heartfelt gratitude to the anonymous reviewers for their valuable insights and constructive feedback, which have significantly enhanced the quality of this article. Their contributions have been instrumental in refining the research and strengthening the overall content. We acknowledge them as anonymous collaborators who has played a vital role in shaping this work.

Zhan Su and Jun Ai would like to express their endless love to Lingyi Ai for being a constant source of inspiration and support, encouraging us to persevere and continue our efforts. We are deeply grateful for her encouragement throughout this journey.

## Author Contributions

**Conceptualization:** Zhan Su, Jun Ai.

**Data curation:** Haochuan Yang.

**Formal analysis:** Haochuan Yang, Jun Ai.

**Funding acquisition:** Zhan Su.

**Project administration:** Zhan Su, Jun Ai.

**Resources:** Zhan Su.

**Software:** Haochuan Yang, Jun Ai.

**Supervision:** Zhan Su.

**Validation:** Haochuan Yang, Jun Ai.

**Visualization:** Haochuan Yang, Jun Ai.

**Writing – original draft:** Haochuan Yang.

**Writing – review & editing:** Zhan Su, Jun Ai.

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
