## [Decision Letter · Decision Letter 0]

14 Jun 2023

PONE-D-22-28030Link prediction in recommender systems based on fuzzy preference vectors and community characteristicsPLOS ONE

Dear Dr. Ai,

Thank you for submitting your manuscript to PLOS ONE. After careful consideration, we feel that it has merit but does not fully meet PLOS ONE’s publication criteria as it currently stands. Therefore, we invite you to submit a revised version of the manuscript that addresses the points raised during the review process.

Please carefully check the Reviewers’ comments and improve the manuscript.

Please also check the language – I suggest using professional proofreading.

We look forward to receiving your revised manuscript.

Kind regards,

Agnieszka Konys, Ph.D.

Academic Editor

PLOS ONE

Journal Requirements:

2. Please note that PLOS ONE has specific guidelines on code sharing for submissions in which author-generated code underpins the findings in the manuscript. In these cases, all author-generated code must be made available without restrictions upon publication of the work. Please review our guidelines at https://journals.plos.org/plosone/s/materials-and-software-sharing#loc-sharing-code and ensure that your code is shared in a way that follows best practice and facilitates reproducibility and reuse. New software must comply with the Open Source Definition.

Reviewers' comments:

Reviewer's Responses to Questions

**Comments to the Author**

1. Is the manuscript technically sound, and do the data support the conclusions?

Reviewer #1: Yes

Reviewer #2: Yes

2. Has the statistical analysis been performed appropriately and rigorously? 

Reviewer #1: Yes

Reviewer #2: Yes

3. Have the authors made all data underlying the findings in their manuscript fully available?

Reviewer #1: No

Reviewer #2: Yes

4. Is the manuscript presented in an intelligible fashion and written in standard English?

Reviewer #1: No

Reviewer #2: Yes

5. Review Comments to the Author

Reviewer #1: 1. The paper should be corrected by a native speaker to improve the English.

2. The following very recent work can be also included in the bibliography:

[1] Su, Z., Zheng, X., Ai, J., Shen, Y., & Zhang, X. (2020). Link prediction in recommender systems based on vector similarity. Physica A: Statistical Mechanics and its Applications, 560, 125154.

3. 2. In experimental results, you may check again the results of DTEC-SCoR algorithm that seems to yield very low values.

However, this method is a state of the art RS method that should perform better results,

so the authors have to check again their implementation and experiments.

4. Please, include comparisons with method [1] that also report results on Netflix dataset.

[1] Su, Z., Zheng, X., Ai, J., Shen, Y., & Zhang, X. (2020). Link prediction in recommender systems based on vector similarity. Physica A: Statistical Mechanics and its Applications, 560, 125154.

5.computational complexity comparisons are missing. The values of comp. times reported on Fig. 12 are too low low concerning the size of the datasets. Please, explain better the experiment and provide the used hardware.

6. Please provide some cases that the proposed method fails and explain the reasons.

7. minor changes

Equ. -> Eq.

Fig -> Fig.

Reviewer #2: The paper is good written and well arranged. Author have to follow the guideline that how to arrange the text in paper.

Author can improve the Proposed Method part of the paper. Remove grammatical mistakes.

6. PLOS authors have the option to publish the peer review history of their article (what does this mean?). If published, this will include your full peer review and any attached files.

Reviewer #1: No

Reviewer #2: No

---

## [Author Response · Author response to Decision Letter 0]

4 Aug 2023

Reviewer: 1

Concern # 1: The paper should be corrected by a native speaker to improve the English.

Author response: We appreciate the valuable feedback provided by the reviewer. 

Author action: In response to the reviewer's suggestion, we have carefully reviewed and corrected the English language in our paper for further consideration. All big changes marked yellow, proofreading doesn’t marked.

Concern # 2: The following very recent work can be also included in the bibliography:

[1] Su, Z., Zheng, X., Ai, J., Shen, Y., & Zhang, X. (2020). Link prediction in recommender systems based on vector similarity. Physica A: Statistical Mechanics and its Applications, 560, 125154.

Author response: Thank you for the advice, the suggested work is actually one of our recent publications. We have added in the revised paper as advised.

Author action: On page 14, we have added the suggested algorithm to the revised manuscript and also added another paper on recent publication by our group (labeled as SRA in experiments). We thank the reviewers for being interested in our research. 

[2] Ai, J., Cai, Y., Su, Z., Zhang, K., Peng, D., & Chen, Q. (2022). Predicting user-item links in recommender systems based on similarity-network resource allocation. Chaos, Solitons & Fractals, 158, 112032.

Concern # 3: In experimental results, you may check again the results of DTEC-SCoR algorithm that seems to yield very low values.

However, this method is a state of the art RS method that should perform better results,

so the authors have to check again their implementation and experiments.

Author response: Thank you for the comment, we have found the bug in our implementation and updated the corresponding results. DTEC-SCoR does very good in most of the experiments. 

Author action: We have rewritten part of DTEC-SCoR implementation, the results is shown in the experiment sections on page 18-23.

Concern # 4: Please, include comparisons with method [1] that also report results on Netflix dataset.

[1] Su, Z., Zheng, X., Ai, J., Shen, Y., & Zhang, X. (2020). Link prediction in recommender systems based on vector similarity. Physica A: Statistical Mechanics and its Applications, 560, 125154.

Author response: Thank you for this comment. The method has been added as the reviewer suggested.

Author action: The updated results can be seen from pages 14 in the revised manuscript. We also replace the older Bhattacharyya CF (2015) with a newer Method SRA (2023).

Concern # 5: Computational complexity comparisons are missing. The values of comp. times reported on Fig. 12 are too low low concerning the size of the datasets. Please, explain better the experiment and provide the used hardware.

Author response: We appreciate the reviewer’s comment and have updated the manuscript accordingly.

Author action: We have added a paragraph to address this issue, which can be found on pages 22-23.

Concern # 6: Please provide some cases that the proposed method fails and explain the reasons.

Author response: Thanks to your suggestion, in the discussion of the algorithm and in the conclusion vignette, we have added a related discussion.

Author action: Corresponding revisions can be seen on pages 23 of the revised paper, where the revisions have been marked yellow. “However,…”

Concern # 7: minor changes

Equ. -> Eq.

Fig -> Fig.

Author response: Thank you for the comment. 

Author action: The relevant abbreviations have been revised. However, the “MANUSCRIPT BODY FORMATTING GUIDELINES” of PLOS ONE suggests that Figures should be cited as “Fig 1” and equation citation should be “Eq 1”. Therefore, we keep them as the journal requires.

 

Reviewer: 2

Concern # 1: The paper is good written and well arranged. Author have to follow the guideline that how to arrange the text in paper.

Author can improve the Proposed Method part of the paper. Remove grammatical mistakes.

Author response: Thanks to the reviewer's encouragement, we revised the paper according to the journal's guideline. It was also proofread for grammar and expression.

Author action: The paper has been revised accordingly, all big changes marked yellow, proofreading doesn’t marked.

---

## [Editor Report · Decision Letter 1]

14 Aug 2023

FPLV: Enhancing Recommender Systems with Fuzzy Preference, Vector Similarity, and User Community for Rating Prediction

PONE-D-22-28030R1

Dear Dr. Ai,

We’re pleased to inform you that your manuscript has been judged scientifically suitable for publication and will be formally accepted for publication once it meets all outstanding technical requirements.

Kind regards,

Agnieszka Konys, Ph.D.

Academic Editor

PLOS ONE
---

## [Editor Report · Acceptance letter]

17 Aug 2023

PONE-D-22-28030R1 

FPLV: Enhancing Recommender Systems with Fuzzy Preference, Vector Similarity, and User Community for Rating Prediction 

Dear Dr. Ai:

I'm pleased to inform you that your manuscript has been deemed suitable for publication in PLOS ONE. Congratulations! Your manuscript is now with our production department. 

Kind regards, 

on behalf of

Dr. Agnieszka Konys 

Academic Editor

PLOS ONE